# Asymmetric synthesis of tetrazole and dihydroisoquinoline derivatives by isocyanide-based multicomponent reactions

Qian Xiong[1], Shunxi Dong[1], Yushuang Chen[1], Xiaohua Liu[1] & Xiaoming Feng[1]

Although isocyanide-based multicomponent reactions were proven to be simple, elegant and facile strategies for the synthesis of highly valuable nitrogen-containing heterocycles, their asymmetric versions accessing to optically active nitrogen heterocyclic compounds are rather limited. Here, we illustrate that, relying on the enantioselective addition of simple isocyanides to C=C bonds, several isocyanide-based multicomponent reactions are realized in the presence of a chiral $Mg^{II}$-$N,N'$-dioxide catalyst. In the reaction among isocyanide, $TMSN_3$, and alkylidene malonate, three- or four-component reactions could be precisely controlled by modulating reaction conditions, supplying two types of enantioenriched tetrazole-derivatives in moderate to high yields. Possible catalytic cycles via a key zwitterionic intermediate, and the vital roles of $H_2O$ or excess ligand are provided based on control experiments. Moreover, taking advantage of this zwitterionic intermediate as a 1,3-dipole, an enantioselective dearomative [3+2] annulation reaction of nonactivated isoquinolines is achieved, furnishing chiral 1,2-dihydroisoquinolines in good to excellent results.

[1] Key Laboratory of Green Chemistry & Technology, Ministry of Education, College of Chemistry, Sichuan University, Chengdu 610064, China. Correspondence and requests for materials should be addressed to X.L. (email: liuxh@scu.edu.cn) or to X.F. (email: xmfeng@scu.edu.cn)

Nitrogen-containing heterocycles are the most popular structural components of pharmaceuticals and agrochemicals[1–3]. Among the vast array of such compounds, tetrazole and hydroisoquinoline motifs are particularly interesting and have been included in top 25 most commonly utilized nitrogen heterocycles in pharmaceuticals[3]. The substances with tetrazole motifs exhibit various types of biological properties (Fig. 1a)[4–7]. For instance, 1,5-disubstituted tetrazoles, as bioisosteres of the cis-amide bond in peptides, display similar physicochemical properties but enhancement of metabolic stability in living systems[7]. Hydroisoquinoline skeleton is widely represented in bioactive alkaloids, such as jamtine, haiderine, and crystamidine (Fig. 1a)[8–10]. In consequence, the development of synthetic approaches to tetrazole and reduced isoquinoline derivatives, especially enantioenriched ones having a variety of peripheral functional groups, would therefore be great of importance for the exploration and modification of new bioactive molecules or drugs. In this regard, great endeavors have been continuously devoted for the synthesis of these valuable compounds over the past two decades[11–14].

Isocyanide-based multicomponent reactions (IMCRs) represent one of the most efficient methods for the construction of heterocycles, wherein a cascade of elementary chemical reactions occur in a uniquely ordered manner under the same reaction conditions, allowing rapid assembly of complex molecules in one pot from simple starting materials with remarkable synthetic efficiency and high atom economy[15–22]. In this context, many IMCRs accessing to 1,5-disubstituted tetrazoles and 1,2-dihydroisoquinolines were reported in the past decade[20–25]. However, in sharp contrast, enantioselective versions of IMCRs are relatively limited and most of studies are focused on the classic Passerini reaction and Ugi reaction initiated by nucleophilic addition of isocyanides to C=X bonds (X=O, or NR')[15,26]. Among the approaches explored to date, asymmetric syntheses of enantioenriched 1,5-disubstituted tetrazoles and 1,2-dihydroisoquinolines are rare. To the best of our knowledge, the only example of catalytic enantioselective synthesis of chiral tetrazoles via Passerini-type reaction was reported in 2008 by Wang and Zhu[27]. In this work, key for success was the use of chiral [(salen) Al^III Me] catalyst with a single coordination site, and moderate to high e.r. values were achieved with aliphatic aldehydes to construct 5-(1-hydroxyalkyl)tetrazoles. Mechanistically, the component of aldehyde in Passerini-type reaction could be replaced by the substrates with polar C=C bond to give a zwitterionic intermediate I (Fig. 1b), which potentially participates in various subsequent multicomponent processes[23]. Although dearomative [3+2] annulation reaction based on this hypothesis would appear to be highly attractive for the synthesis of chiral 1,2-dihydroisoquinolines, the asymmetric version of this reaction remains elusive[28–30]. Intrigued by the above hypothesis and good performance of N,N'-dioxide-based chiral catalysts showed in enantioselective addition to α,β-unsaturated carbonyl compounds[31–34], we envisioned that our chiral N,N'-dioxide/metal complex could probably be suitable to catalyze the kinds of multicomponent reactions mentioned above. Herein, we describe the asymmetric IMCRs initiated by the enantioselective addition of simple isocyanides to olefins (Fig. 1b). Interestingly, through adjusting reaction conditions, we are able to accurately control the number (1 or 2)[35-38] of isocyanide molecules participated in the reaction of alkylidene malonates, isocyanides, and TMSN₃, leading to two types of enantioenriched tetrazole derivatives 4 and 5, respectively, in moderate to high yields. Moreover, taking advantage of the same zwitterionic intermediate I as a 1,3-dipole, an enantioselective formal [3+2] cycloaddition reaction with isoquinolines is also realized, furnishing the corresponding 1,2-dihydroisoquinolines in good to excellent results. This catalytic protocol also represents a rare example of asymmetric direct dearomatization of nonactivated isoquinoline[39].

## Results

**Optimization of reaction conditions.** Our initial attempts involved using dimethyl 2-(cyclohexylmethylene) malonate as the model substrate together with 2-naphthyl isocyanide, TMSN₃ to optimize the reaction conditions. First, the investigation of metal salts was carried out in the presence of chiral N,N'-dioxide ligand **L-PrPr₂**. It was found that only Mg(OTf)₂/**L-PrPr₂** complex could promote the reaction smoothly, affording two different 1,5-disubstituted tetrazole derivatives. One was assigned to be compound **4a** via one molecular isocyanide-involved three-component reaction (29% yield, 82:18 e.r.) and the other was compound **5a** incorporated bimolecular isonitrile via a four-component reaction (53% yield, 92.5:7.5 e.r. For details, see Supplementary Figs. 1–7). But messy mixture was obtained with metal salts as Sc (OTf)₃ (for details, see Supplementary Table 1). Encouraged by these results, various chiral N,N'-dioxide ligands coordinating with Mg(OTf)₂ were evaluated (Table 1, entries 1–4). Chiral N,N'-dioxide ligands had a significant effect on the distribution of the products and e.r. values. When employing L-pipecolic acid-derived **L-PiPr₂** as the ligand, interestingly, only four-component reaction product **5a** was obtained in 97% yield with lower e.r. value (entry 2; 82:18 e.r. vs 92.5:7.5 e.r.). Changing the chiral backbone of the ligand to L-ramipril acid-derived **L-RaPr₂**, higher e.r. values were achieved for both product **5a** (46% yield, 96:4 e.r. vs 92.5:7.5 e.r.) and the product **4a** (38% yield, 83.5:16.5 e.r. vs 82:18 e.r. For details, see Supplementary Table 2). No better results were afforded with further increase of the steric hindrance of aniline in ligand (entry 4). Next, other parameters were systematically screened. Noteworthily, the ratio of Mg(OTf)₂ to **L-RaPr₂** displayed a considerable influence on the product distribution (entries 5–6). With excessive amount of metal salt (Mg^II/**L-RaPr₂** = 1.5/1.0), four-component product **5a** was formed dominantly in 88% yield with 94:6 e.r. (entry 5). On the contrary, three-component product **4a** prevails (entry 6; 53% yield with 88:12 e.r. for **4a**, and 25% yield with 95.5:4.5 e.r. for **5a**) with increasing the amount of **L-RaPr₂** (Mg^II/**L-RaPr₂** = 1.0/1.5). After fine-tuning, the optimized conditions for four-component reaction were set to be Mg(OTf)₂/**L-RaPr₂** (1.4/1.0, 10 mol%) as the catalyst in CH₂ClCH₂Cl at 30 °C for 3 h, and the desired product **5a** was provided in 91% yield with 94.5:5.5 e.r. (conditions A, entry 7). To our delight, it was found that four-component reaction was almost suppressed at lower reaction temperature (−40 °C), and the three-component product **4a** was isolated as the main product (entry 8; 57% yield with 94.5:5.5 e.r.). However, the reactivity dropped dramatically and extending the reaction time was needed to get moderate yield (entry 9; 73% yield). Fortunately, the addition of 5 Å molecular sieve could accelerate the reaction without any impact on the enantioselectivity (entry 10; 91% yield with 95:5 e.r.). After adjustment of temperature and ratio of components, the optimized conditions for three-component product **4a** entailed the use of the Mg(OTf)₂/**L-RaPr₂** (1.0/1.5, 10 mol%) as the catalyst and 5 Å MS as the additive in CH₂Cl₂ at −40 °C for 2 days, then −20 °C for 3 days (conditions B, entry 10).

**Substrate scope of three-component tetrazoles.** With the standard reaction conditions in hand (Table 1, entry 10), the substrate scope of three-component reaction was explored. As depicted in Fig. 2, a wide range of alkylidene malonates were investigated. We first examined the effect of the ester moiety. Aside from dimethyl 2-(cyclohexylmethylene)malonate, diethyl, diisopropyl or dibenzyl 2-(cyclohexylmethylene)malonates were also suitable

**Fig. 1** Representative active substances and our strategies based on IMCRs for their construction. **a** Active substances with tetrazole and hydroisoquinoline motifs. **b** Our strategies to asymmetric synthesis of tetrazole and dihydroisoquinoline derivatives by IMCRs

substrates, delivering the corresponding products **4a**–**4d** with equally good enantioselectivities (95:5–97:3 e.r.). Nevertheless, the reactivity gradually decreased with the increase of steric hindrance of the ester group (52–91%). It was noteworthy that the size of ring of β-cyclic alkyl substituents showed an apparent influence on the reactivities and enantioselectivities (**4e**–**4h**). Of the rings investigated, seven-membered one led to the lowest yield (22%) and excellent enantioselectivity (96:4 e.r.). Raising the reaction temperature to −10 °C, a higher yield (67%) with maintained enantioselectivity was achieved for the product **4e**. With the decrease of ring size from five-membered to three-membered, both yields and e.r. values of the products **4f**–**4h** decreased remarkably. The substrates derived from isobutyl aldehyde, were found to be applicable in this reaction, yielding the corresponding products **4i** and **4j** in good yields (72 and 84%) and high enantioselectivities (96:4 e.r.). We also tested with other linear or branched aliphatic substituted ones, and the corresponding products **4k**–**4m** were afforded in good to excellent yields (56–93%) with slightly lower enantioselectivities

(86:14–88.5:11.5 e.r.). Next, the tolerance of isocyanide in three-component reaction was examined. Compared with 2-naphthyl isocyanide **1a**, the reaction of 4-nitrophenyl isocyanide could also furnish the target product (**4n**) in medium yield (50%) and good enantioselectivity (90.5:9.5 e.r.) at 0 °C.

Initial attempt of using aromatic aldehyde-derived methylene malonates in the reaction system was unsuccessful and three-component product was obtained in extremely lower yield (for details, see Supplementary Table 5). It was interesting that the addition of water not only accelerated the rate of reaction but also suppressed four-component reaction process. Thus, different aryl-substituted alkylidene malonates were subjected to the reaction. The positions of substituent at aromatic ring of alkylidene malonate also significantly affected the outcomes of the reaction. *Para*-substituted substrates gave the desired products (**4p**, **4r**, and **4s**) with both higher yields and enantioselectivities than those of *ortho*- or *meta*-substituted ones (**4q**, **4t**, and **4u**). Fused-ring-substituted alkylidene malonate provided the corresponding product (**4v**) in 39% yield and 88:12

**Table 1 Optimization of the reaction conditions**

L-PrPr₂: R = 2,6-$^i$Pr₂C₆H₃, n = 1
L-PiPr₂: R = 2,6-$^i$Pr₂C₆H₃, n = 2

L-RaPr₂: R = 2,6-$^i$Pr₂C₆H₃
L-RaPr₃: R = 2,4,6-$^i$Pr₃C₆H₂

| entry | Ligand (L) | Mg(OTf)₂: L | T (°C) | 4a Yield (%)[a] | 4a e.r.[b] | 5a Yield (%)[a] | 5a e.r.[b] |
|---|---|---|---|---|---|---|---|
| 1 | **L-PrPr₂** | 1.0:1.0 | 35 | 29 | 82:18 | 53 | 92.5:7.5 |
| 2 | **L-PiPr₂** | 1.0:1.0 | 35 | – | – | 97 | 82:18 |
| 3 | **L-RaPr₂** | 1.0:1.0 | 35 | 38 | 83.5:16.5 | 46 | 96:4 |
| 4 | **L-RaPr₃** | 1.0:1.0 | 35 | 36 | 72:28 | 43 | 86.5:13.5 |
| 5 | **L-RaPr₂** | 1.5:1.0 | 35 | – | – | 88 | 94:6 |
| 6 | **L-RaPr₂** | 1.0:1.5 | 35 | 53 | 88:12 | 25 | 95.5:4.5 |
| 7[c] | **L-RaPr₂** | 1.4:1.0 | 30 | – | – | 91 | 94.5:5.5 |
| 8[d] | **L-RaPr₂** | 1.0:1.5 | −40 | 57 | 94.5:5.5 | – | – |
| 9[d,e] | **L-RaPr₂** | 1.0:1.5 | −40 | 73 | 95:5 | – | – |
| 10[f] | **L-RaPr₂** | 1.0:1.5 | −40 then −20 | 91 | 95:5 | – | – |

Unless otherwise noted, all reactions were carried out with **1a** (0.10 mmol), **2a** (0.12 mmol), **3a** (0.10 mmol), and Mg(OTf)₂/**L-RaPr₂** (1.0:1.0, 10 mol%) in CH₂Cl₂ (1.0 mL) at 35 °C for 3 h
[a]Isolated yield
[b]Determined by CSP-HPLC analysis
[c]Carried out with **1a** (0.20 mmol), **2a** (0.24 mmol), **3a** (0.20 mmol), and Mg(OTf)₂/**L-RaPr₂** (1.4:1.0, 10 mol%) in CH₂ClCH₂Cl (1.0 mL) at 30 °C for 3 h
[d]−40 °C for 7 days
[e]With 5 Å MS (10 mg) as the additive
[f]Carried out with **1a** (0.10 mmol), **2a** (0.15 mmol), **3a** (0.15 mmol), 5 Å MS (10 mg), and Mg(OTf)₂/**L-RaPr₂** (1.0/1.5, 10 mol%) in CH₂Cl₂ (1.0 mL) at −40 °C for 2 days, then −20 °C for 3 days

e.r. (−)-Citronellal-derived substrate yielded the corresponding product **4w** in 47% yield and 2.8:1 d.r. at −10 °C. The absolute configuration of the product **4i** was explicitly determined to be *S*-configuration by single-crystal X-ray diffraction analysis.

**Substrate scope of four-component tetrazoles**. Having established the optimized conditions for the synthesis of four-component products (Table 1, entry 7), we evaluated the generality of this transformation (Fig. 3). The ring size of alkylidene malonates appeared to have a similar influence on the enantioselectivities with that of three-component reaction, and the substrates with smaller rings were prone to give lower enantioselectivities (**5a–5d**). Other branched aldehyde-derived substrates were suitable as well to afford the desired product **5e–5h** in high yields (92–99% yield) with moderate to good e.r. values (80:20–93:7 e.r.). In addition, the ester group in alkylidene malonates could be extended to others (**5i–5l**). Highest e.r. value was obtained with Bn group. It should be pointed out that the aromatic aldehyde-derived substrates were applicable, delivering the corresponding products (**5o–5x**) in good yields, however, the enantioselectivities depended on the electronic nature and position of the substituent on aryl group of the alkylidene malonates. Higher enantioselectivities were observed with substrates containing electron-withdrawing groups than those containing electron-donating ones (**5o–5v**). Similarly, compared with the

*meta-* and *ortho*-substituted substrates, *para*-substituted ones gave the desired products with higher enantioselectivities (**5r–5u**). Naphthyl substituted alkylidene malonate was also tolerated to yield the product (**5w**) in good results. Regarding to (−)-Citronellal-derived substrate, higher diastereoselectivity was obtained with *ent*-**L-RaPr₂** as ligand (1:4.0 d.r. vs 2.0:1 d.r.). Notably, the screening of isocyanides indicated that the electronic properties of substituents on aromatic rings of isocyanide displayed a profound effect on the course of the reaction (shown in **5l–5n**), and replacing the electron-donating substituents with the electron-withdrawing group at *para*-position of the phenyl ring, for example, nitro group, only three-component product was detected rather than the desired four-component product. The structure of the racemic adduct s-*trans*-**5a** was confirmed by single-crystal X-ray diffraction analysis. Next, derivatization of the product **5e** was conducted. One of the ester groups could be easily removed in the presence of LiCl and H₂O with maintained enantioselectivity (**6e**; 70% yield, 85:15 e.r.). It is worth noting that imide moiety was very stable even under high temperature.

**Substrate scope of 1,2-dihydroisoquinoline derivatives**. As illustrated in Fig. 1b, the nucleophilic addition of isocyanides to polar C=X or C=C bonds affords zwitterionic intermediates, which theoretically could be used as 1,3-dipoles. However, comparing with classic Passerini and Ugi reactions, where these

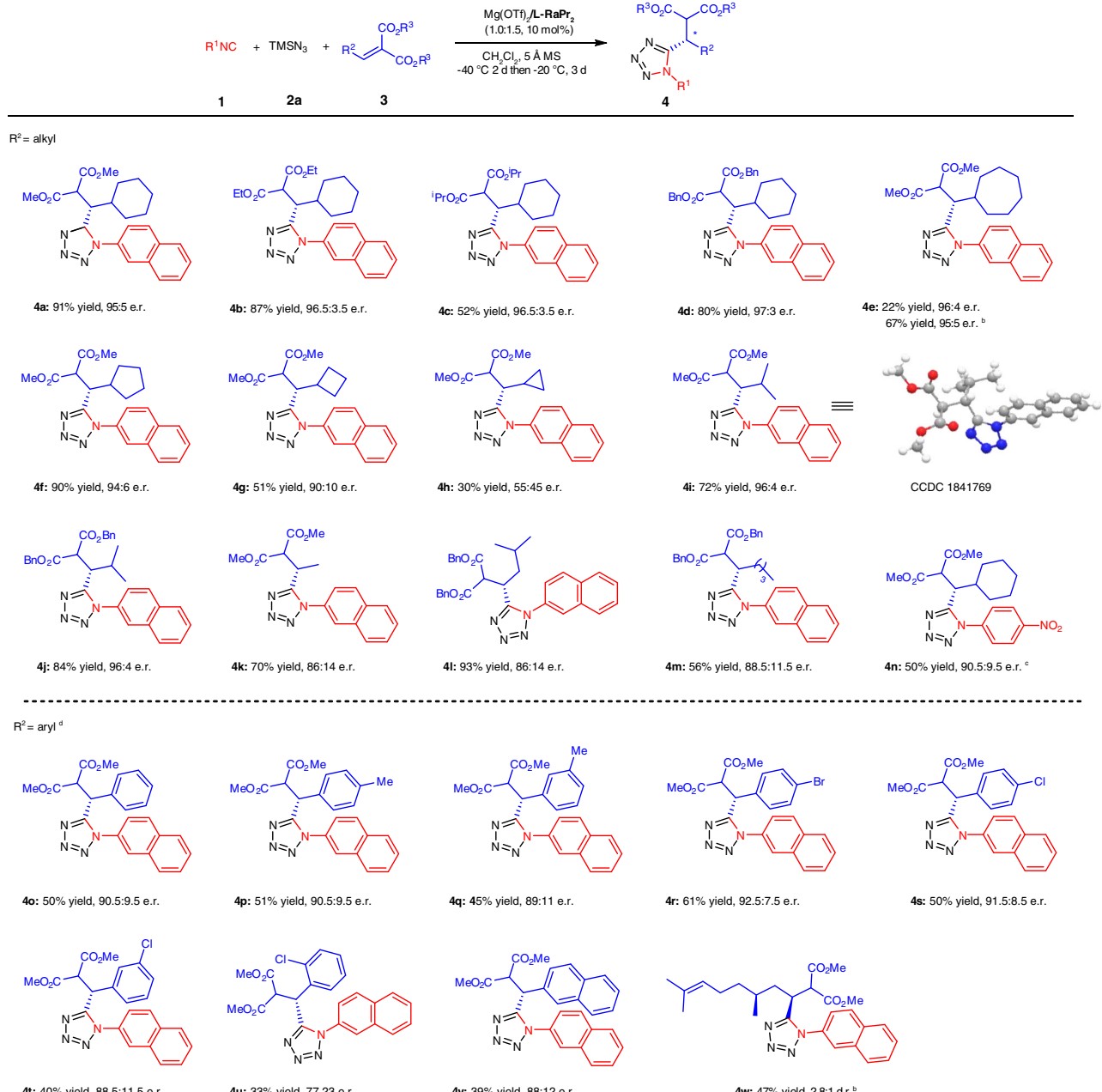

**Fig. 2** Substrate scope of asymmetric synthesis three-component tetrazoles. Unless otherwise noted, all reactions were carried out with **1** (0.10 mmol), **2a** (0.15 mmol), **3** (0.15 mmol), 5 Å MS (10 mg), and Mg(OTf)$_2$/**L-RaPr$_2$** (1.0/1.5, 10 mol%) in CH$_2$Cl$_2$ (1.0 mL) at −40 °C for 2 days, then −20 °C for 3 days. Isolated yield and e.r. was determined by CSP-HPLC analysis. [b]Performed at −10 °C. [c]Performed at 0 °C with Mg(OTf)$_2$/**L-RaPr$_2$** (1.0/1.0, 10 mol%). [d]Carried out with **1** (0.10 mmol), **2a** (0.15 mmol), **3** (0.10 mmol), H$_2$O (5 μL), and Mg(OTf)$_2$/**L-RaPr$_2$** (1.0/1.0, 10 mol%) in CH$_2$Cl$_2$ (1.0 mL) at 30 °C for 2 days

zwitterionic intermediates were captured by carboxylic acid, the utility of these species in cycloaddition reactions is rather limited. To the best of our knowledge, only sporadic racemic versions of such reactions were reported to date[23–25]. Encouraged by the good performance of our chiral Mg$^{II}$-N,N′-dioxide Lewis acid catalyst for the synthesis of 1,5-disubstituted tetrazole derivatives, we assumed if the zwitterionic intermediate **I** was captured by imine, another hetereo-cyclic amidine-type of product could be obtained. Many kinds of combinations with different electrophiles, isocyanides, and dipolarophiles were tested. The trick is how to control a cascade of elementary chemical reactions occurring in a uniquely ordered manner, since two reactants are electrophiles. It should be a better choice if the reactivity of

dipolarophile is lower than the electrophile for the isocyanide-initiated addition step. Finally, phenyl-substituted alkylidene malonate **3o**, isoquinoline **7a**, and *tert*-butyl isocyanide **1e** were employed in the model reaction to optimize the reaction conditions. We are glad to find that by slight modification the four-component reaction conditions as Mg(OTf)$_2$/**L-RaPr$_2$** (1.2/1.0, 10 mol%) in CH$_2$ClCH$_2$Cl (0.5 mL) at 35 °C for 2 days, the desired cycloaddition product **8a** was produced as a single diastereomer in 85% yield with 96.5:3.5 e.r. (for details, see Supplementary Tables 7–9). Under such conditions, the substrates with respect to alkylidene malonates were evaluated. As shown in Fig. 4, a broad range of alkylidene malonates bearing different ester groups or β-aryl substituents reacted smoothly to deliver the

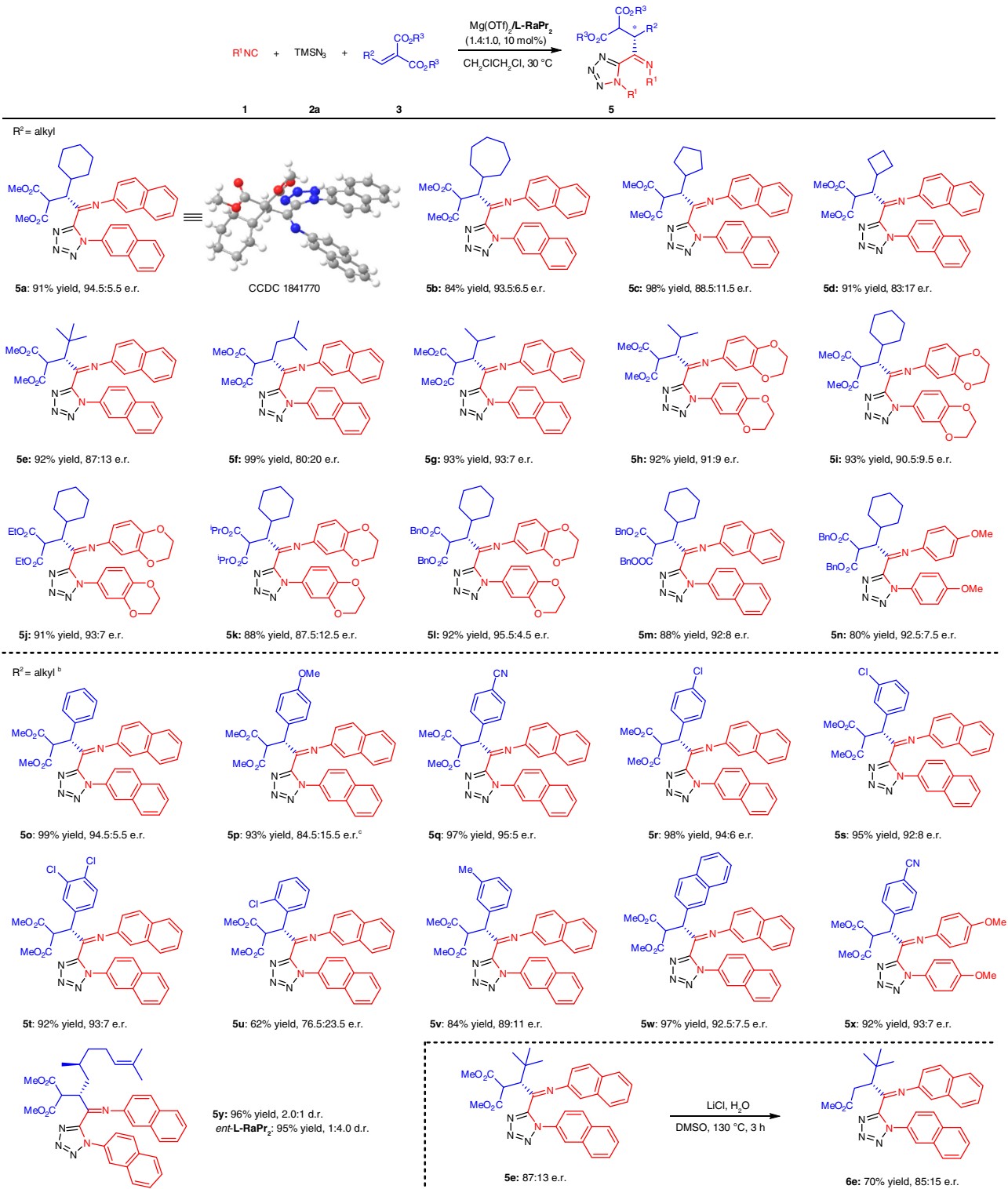

**Fig. 3** Substrate scope of asymmetric synthesis four-component tetrazoles. Unless otherwise noted, all reactions were carried out with **1** (0.20 mmol), **2a** (0.24 mmol), **3** (0.20 mmol), and Mg(OTf)$_2$/**L-RaPr$_2$** (1.4/1.0, 10 mol%) in CH$_2$ClCH$_2$Cl (1.0 mL) at 30 °C for 3 h. Isolated yield and e.r. was determined by CSP-HPLC analysis. [b]Carried out with **1** (0.20 mmol), **2a** (0.20 mmol), **3** (0.15 mmol). [c]20 mol% of NaBAr$^F_4$ {sodium tetrakis[3,5-bis(trifluoromethyl)phenyl]borate} was added

corresponding products with good yields and high enantioselectivities (**8a–8h**; 79–93% yield, 92:8–97:3 e.r.). Comparatively, aliphatic aldehyde-derived alkylidene malonate was less efficient, yielding **8t** in much lower yield and enantioselectivity (21% yield, 77:23 e.r.). On the other hand, other substituted isoquinolines

were suitable as well regardless of electronic nature of the substitutions. The desired cycloaddition products (**8i–8p**) were isolated in excellent enantioselectivities (96:4–99:1 e.r.) with slightly lower yields (40–72%). Next, the screening of isocyanides implied that the steric hindrance exhibited a considerable effect on the e.r.

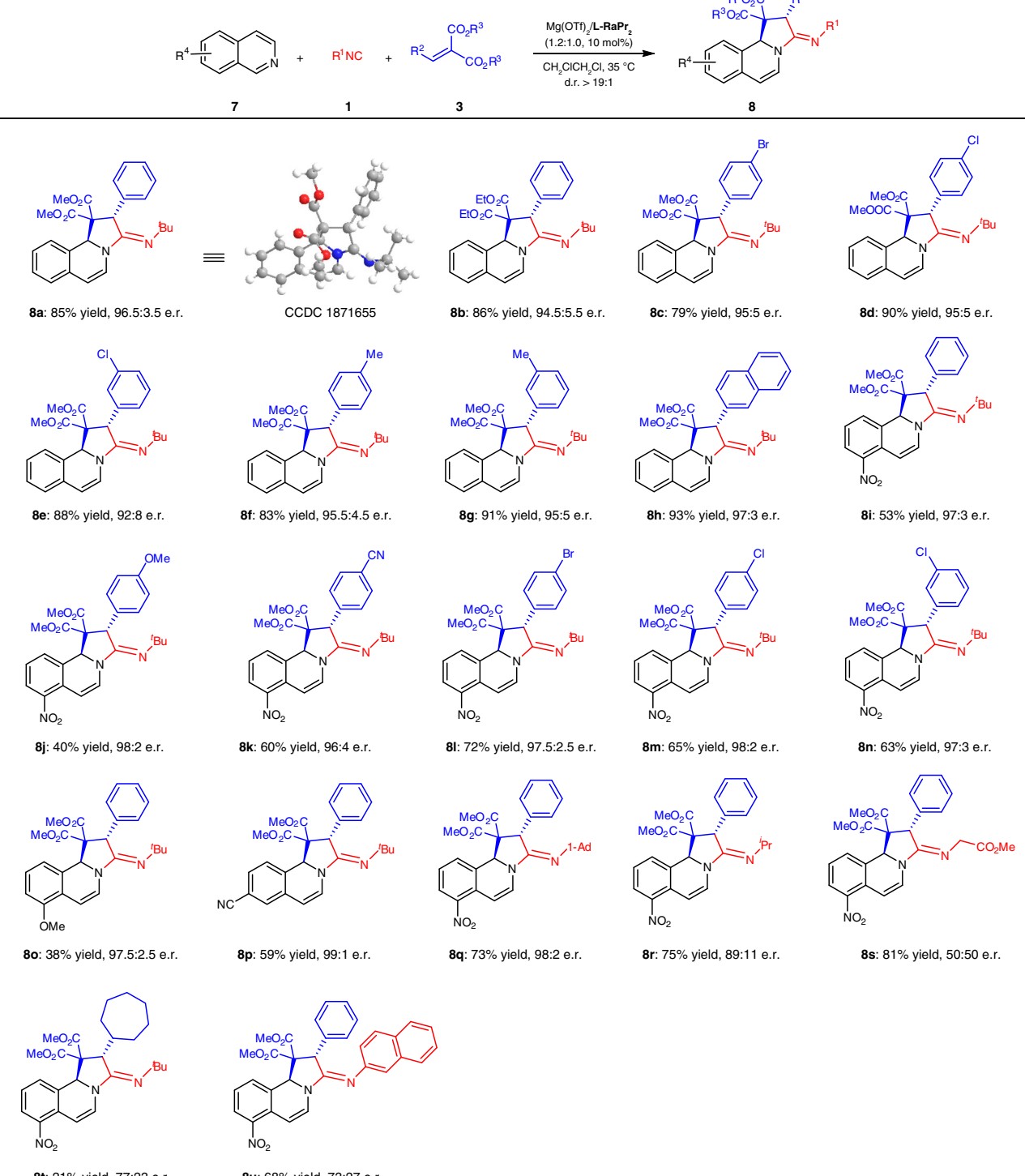

**Fig. 4** Substrate scope of asymmetric synthesis 1,2-dihydroisoquinoline derivatives. All reactions were carried out **7** (0.10 mmol), **1** (0.15 mmol), **3** (0.15 mmol), and Mg(OTf)$_2$/**L-RaPr$_2$** (1.2:1.0, 10 mol%) in CH$_2$ClCH$_2$Cl (0.5 mL) at 35 °C for 2 days. Isolated yield and e.r. was determined by CSP-HPLC analysis

values of the product; the more bulk 1-adamantanyl isocyanide gave 98:2 e.r., while the less bulk isocyanides afford the product (**8r** and **8s**) with much lower enantioselectivities. In addition, switching to aromatic isocyanide resulted in a rapid drop of e.r. value (**8r**). The absolute configuration of the product **8a** was determined to be (2S, 10bS, E) by single-crystal X-ray diffraction analysis. It should be noted that this transformation was a simultaneous dearomative process of isoquinolines, delivering

fused polycyclic 1,2-dihydroisoquinoline-based amidine derivatives with a single diastereomer in all cases.

**Control experiments and proposed catalytic cycle.** In order to get insight into the reaction mechanism, we carried out several control experiments. The first one was deuterium experiment under four-component reaction conditions (Table 1, entry 7) to understand the source of proton. As reflected in Fig. 5a, when

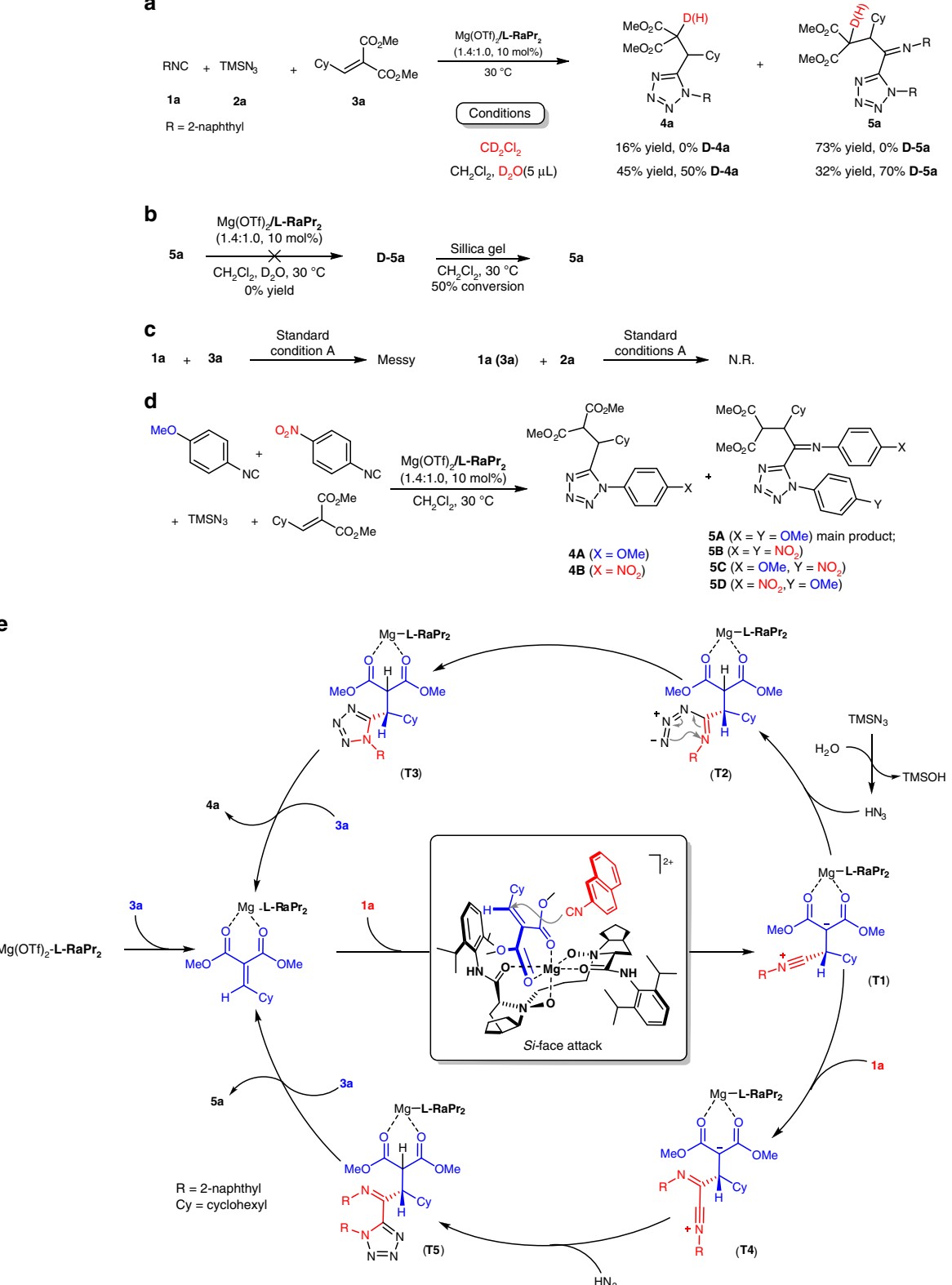

**Fig. 5** Control experiments and proposed catalytic cycle. **a** Deuterium labeling experiments. **b** Control experiments for **5a** and **D-5a**. **c** Control experiments for two materials. **d** NMR spectra and HRMS for reaction involving two different isocyanides. **e** Proposed catalytic cycle for IMCRs

deuterated dichloromethane was used as the solvent, the four-component product **5a** without deuterium was obtained as the major product (**4a**: 16% yield, **5a**: 73% yield). While the addition of a small amount of $D_2O$ (5 μL) dramatically affected the ratio of two products. In this case, three-component product **4a** (with

50% deuterium) was isolated as the main product instead of **5a** (with 70% deuterium). In addition, the percent of deuterated **5a** decreased from 70 to 35% by treatment with silicone gel (Fig. 5b). These results not only indicated that the proton came from a trace amount of water or work up with silicone gel but also

highlighted the remarkable influence of water on the pathway. With this observation, $HN_3$ was supposed to be involved in both three-component reaction pathway and four-component reaction pathway. Next, we evaluated the possible transformations of each two reactants. It showed that in the presence of a chiral Lewis acid catalyst, no reaction occurred from $TMSN_3$ **2a** with isocyanide **1a** or alkylidene malonate **3a**. Isocyanide **1a** could react with **3a** rapidly, generating a messy mixture. These results implied that the reaction is triggered by the addition of isocyanide **1a** to **3a** catalyzed by chiral Lewis acid. Furthermore, we subjected an equal amount of two electron-different isocyanides for the multicomponent reaction, and the characteristic signal of main product **5A** was detected by NMR analysis and the molecular ion peaks of other products **4A**, **4B**, **5B**, **5C**, and **5D** can be detected by high resolution time of flight mass spectrometry (Fig. 5c). The cross-reaction products **5C** and **5D** confirmed again that the intermediate generated from the addition of isocyanide to alkylidene involved in the initial step of both three-component reaction and four-component reaction.

Based on the control experiments and the previous works[27,40,41], a possible catalytic cycle along with a transition-state model were proposed to elucidate the reaction processes and the chiral induction. As depicted in Fig. 5e, at the outset of the reaction, the *N,N'*-dioxide **L-RaPr₂** and the bidentate alkylidene malonate **3a** coordinated to the $Mg^{II}$ center to form a complex with octahedral geometry. Since the *Re* face of alkylidene malonate **3a** was shielded by the bulk neighboring amide group of the ligand, 2-naphthyl isocyanide **1a** attacked the *Si* face of **3a** to form the intermediate **T1**, which could undergo two different pathways. One is three-component reaction process, in which the intermediate **T1** was trapped $HN_3$ formed in situ[42] to give the intermediate **T2**, which subsequently underwent cyclization to afford **T3**. Then, the catalyst released from the product **4a** to close this catalytic cycle. The other four-component process started from the intermediate **T1**, trapping by another molecule of 2-naphthyl isocyanide **1a** to afford the zwitterionic Intermediate **T4**, which was subsequently reacted with $TMSN_3$ or $HN_3$ in a similar manner to yield the corresponding tetrazole product **5a**.

As showing in Fig. 5e, the reaction pathways were determined by the trap of the intermediate **T1**, with either $N_3^-$ or another molecule of isocyanide. Regarding our experimental outcomes, several factors were found to be crucial. The first one is the electronic nature of isocyanides. According to the observations mentioned in Figs. 2 and 3, electron-rich isocyanides with higher nucleophilicity were prone to underwent four-component reaction, whereas electron-poor isocyanides is favorable for three-component reaction. The reaction conditions were also critical. The addition of either *N,N'*-dioxide **L-RaPr₂** or water is conducive to three-component reaction. In this process, excess amount of the *N,N'*-dioxide **L-RaPr₂** could serve as a Lewis base to activated $TMSN_3$ by coordination, which facilitated the nucleophilic addition of $N_3^-$ to intermediate **T1**. In addition, in the presence of water, $HN_3$ was formed in situ, which was proved to be an efficient strategy in Passerini tetrazole reaction[27,43]. Regarding to excessive metal-promoted four-component reaction pathway, we also screened $Mg^{II}$ with other counteranions ($ClO_4^-$ or $NTf^-$) or other metal salts [LiOTf, NaOTf, and $Ca(OTf)_2$], it seemed that $OTf^-$ anion was beneficial to four-component reaction[37].

## Discussion
Triggered by the enantioselective addition of isocyanides to C=C bonds, several enantioselective isocyanide-based multicomponent reactions were realized by using chiral $Mg^{II}$-*N,N'*-dioxide Lewis acid catalyst, providing a direct and efficient access to highly valuable chiral 1,5-disubstituted tetrazoles and 1,2-

dihydroisoquinolines. On one hand, we accomplished the asymmetric MCRs of isocyanides, $TMSN_3$ with alkylidene malonates. With modification of reaction conditions, both one molecule of isocyanide insertion process and two molecules of isocyanide incorporated pathway could be accurately regulated, furnishing two types of enantioenriched 1,5-disubstituted tetrazoles with good outcomes. On the other hand, the zwitterionic intermediate resulted from the addition of isocyanides to C=C bonds was successfully employed as a 1,3-dipole in the example of enantioselective dearomative [3+2] annulation reaction of isoquinolines, affording chiral fused polycyclic 1,2-dihydroisoquinoline-based amidine derivatives in good to excellent results (up to 93% yield, >19:1 d.r., 99:1 e.r.). It is expected that our findings in this work could stimulate further exploration in asymmetric IMCRs. Further studies on applying this catalyst system to other IMCRs are currently underway in our lab.

## Methods
**General procedure for asymmetric alkyl substituted three-component tetrazoles**. To an oven-dried tube under nitrogen atmosphere were added $Mg(OTf)_2$ (0.010 mmol, 10 mol%), **L-RaPr₂** (0.015 mmol, 15 mol%), 2-(cyclohexylmethylene) malonate **3a** (0.15 mmol), 5 Å MS (10 mg), and $CH_2Cl_2$ (1.0 mL). The mixture was stirred in $CH_2Cl_2$ (1.0 mL) at 35 °C for 0.5 h. Subsequently, $TMSN_3$ **2a** (0.15 mmol) and 2-naphthyl isocyanide **1a** (0.10 mmol) were added at −40 °C. The mixture was stirred at −40 °C for 2 days then at −20 °C for 3 days, and directly purified by flash chromatography on silica gel (eluent: petroleum ether/ethyl acetate = 9/1) to afford the desired product **4a** (91% yield, 95:5 e.r.).

**General procedure for asymmetric aryl substituted three-component tetrazoles**. To an oven-dried tube under nitrogen atmosphere were added $Mg(OTf)_2$ (0.010 mmol, 10 mol%), **L-RaPr₂** (0.010 mmol, 10 mol%), dimethyl 2-benzylidenemalonate **3o** (0.10 mmol), 5 μL $H_2O$, and $CH_2Cl_2$ (1.0 mL). The mixture was stirred in $CH_2Cl_2$ (1.0 mL) at 35 °C for 0.5 h. Subsequently, $TMSN_3$ **2a** (0.15 mmol) and 2-naphthyl isocyanide **1a** (0.10 mmol) were added at 30 °C. The mixture was stirred at 30 °C for 48 h, and directly purified by flash chromatography on silica gel (eluent: petroleum ether/ethyl acetate = 9/1) to afford the desired product **4o** (50% yield, 90.5:9.5 e.r.).

**General procedure for asymmetric four-component tetrazoles**. To an oven-dried tube under nitrogen atmosphere were added $Mg(OTf)_2$ (0.014 mmol, 14 mol%), **L-RaPr₂** (0.010 mmol, 10 mol%), 2-(cyclohexylmethylene)malonate **3a** (0.20 mmol), and $CH_2ClCH_2Cl$ (1.0 mL). The mixture were stirred in $CH_2ClCH_2Cl$ (1.0 mL) at 35 °C for 0.5 h. Subsequently, $TMSN_3$ **2a** (0.24 mmol) and 2-naphthyl isocyanide **1a** (0.10 mmol) were added at 30 °C. The mixture was stirred at 30 °C for 3 h, and directly purified by flash chromatography on silica gel (eluent: petroleum ether/ethyl acetate = 9/1) to afford the desired product **5a** (91% yield, 94.5:5.5 e.r.).

**General procedure for asymmetric 1,2-dihydroisoquinoline derivatives**. To an oven-dried tube were added $Mg(OTf)_2$ (0.012 mmol, 12 mol%), **L-RaPr₂** (0.010 mmol, 10 mol%), dimethyl 2-benzylidenemalonate **3o** (0.15 mmol), and $CH_2ClCH_2Cl$ (0.5 mL). The mixture was stirred in $CH_2ClCH_2Cl$ (0.5 mL) at 35 °C for 0.5 h. Subsequently, isoquinoline **6a** (0.10 mmol) and tert-butyl isocyanide **7a** (0.15 mmol) were added at 35 °C. The mixture was stirred at 35 °C for 48 h, and directly purified by flash chromatography on silica gel (eluent: petroleum ether/ethyl acetate = 9/1) to afford the desired product **8a** (85% yield, 96.5:3.5 e.r.).

## Data availability
The X-ray crystallographic coordinates for structures **4i**, **5a**, and **8a** reported in this study have been deposited at the Cambridge Crystallographic Data Centre (CCDC), under deposition numbers 1841769, 1841770, and 1871655. These data can be obtained free of charge from The Cambridge Crystallographic Data Centre via www.ccdc.cam.ac.uk/data_request/cif. All other data are available from the corresponding author upon reasonable request.

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

## Acknowledgements

The authors acknowledgements financial support from the National Natural Science Foundation of China (grant nos. 21890723, 21772127, and 21625205), and the National Program for Support of Top-Notch Young Professionals, and the Fundamental Research Funds for the Central Universities.

## Author contributions

Q.X. performed the experiments. Y.S.C. repeated data. S.X.D. participated in structure characterization and discussion. X.M.F. and X.H.L. supervised the project. X.M.F., X.H.L., S.X.D., and Q.X. co-wrote the manuscript.

## Additional information

**Competing interests:** The authors declare no competing interests.

