## [Peer Review File · Nature Communications]

REVIEWERS' COMMENTS:

Reviewer #1 (Remarks to the Author):

In this manuscript, Liu, Feng and coworkers report an interesting isocyanide-based asymmetric multicomponent reaction that allows for the synthesis of nitrogen-embedded heterocycles such as tetrazoles and 1,2-dihydroisoquinolines with high enantiomeric purity. Critical to the success is the unprecedented initiation involving chiral MgII-N,N'-dioxide catalyst-enabled enantioselective addition of simple isocyanide to alkenes. By judicious adjustment of the reaction parameters, multicomponent reactions of isocyanides, TMSN₃ with alkylidene could be tuned in either three-component or four-component fashion. Moreover, the dearomative [3+2] cycloaddition of nonactivated isoquinolines could be achieved based on the same Zwitterionic intermediate for the first time. All these reactions constitute the extremely rare successful examples in controlling the enantioselectivity for multicomponent reactions. Mechanistic studies also well supported the proposed catalytic cycle. In addition, the SI is prepared in high quality.

Therefore, this reviewer suggests the publication of these highly interesting results in Nat. Commun. after the following minor points being addressed.

- 1) Abstract, line 3: "relaying on..." should be changed to "relying on..."
- 2) Highlight of Scheme 1: "High efficient with..." should be changed to "Highly efficient with..."
- 3) In the footnote of Table 1, 1 and 3 should be changed to 1a, 3a in ref g.
- 4) Page 5, the last sentence: "claissic Passerini and Ugi..." should be changed to "classic Passerini and Ugi..."
- 5) Page 6, footnote of Table 4: "All reaction were carried out 7..." should be changed to "All reactions were carried out with 7..."
- 6) Page 7, line 8, left column: "23% yield" should be changed to "21% yield" (according to Table 4)?
- 7) Page 8, line 23, left column: "equivale" should be changed to "equivalent".

Reviewer #2 (Remarks to the Author):

Through the work described in this manuscript, Liu, Feng and co-workers have solved a very important and challenging problem in asymmetric catalysis. As the authors rightly discussed in the introduction, development of catalytic enantioselective variant of isocyanide-based multicomponent reactions has remained a prominent challenge and only a handful of reports exist till date.

Here, the authors have devised a very smart strategy to generate a common intermediate through enantioselective conjugate addition of isocyanides to alkylidene malonates using magnesium-complex of the versatile N-oxide ligand developed in their own lab. The common intermediate is then utilized for three different reactions. In the first two of these reactions, azide addition followed by annulation take place to form a tetrazole. An impressive aspect is that, simply by modifying the reaction conditions, the involvement of either one or two molecules of isocyanide in this reaction can be controlled. In the last reaction, the common intermediate acts as a 1,3-dipole and participates in a formal [3+2]-cycloaddition reaction with isoquinoline.

While the substrate scope is quite wide, the enantioselectivities are not excellent in all the cases. However, this can be ignored considering the challenge at hand and the novelty of the strategy.

With the help of a number of control experiments, a reasonable catalytic cycle is proposed along with a stereochemical model to explain the observed stereoselectivity.

Overall, the work reported in this manuscript makes a very important contribution to the field of organic synthesis. Therefore, I would like to recommend the publication of this manuscript in Nature Communications.

Following minor issues should be sorted out before the manuscript is accepted for publication:

- (1) It would be nice to have a couple of applications and synthetic elaborations of the products.
- (2) In a few places in the first page, 'participate' is mistakenly written as 'anticipate'. These and similar mistakes should be corrected.

Reviewer #3 (Remarks to the Author):

The significance of this manuscript is captured in Scheme 2. Coordination of the alkylidene-malonate by an appropriate chiral-ligated Lewis acid activates the substrate for highly enantioselective Michael addition by isocyanides to produce an intermediate that is now capable of multiple transformations. The authors have identified three transformations that incorporate this intermediate - (1) reaction with TMS azide to form chiral tetrazoles, (2) reaction with isoquinolines to produce chiral heterocycles by dearomatization, and (3) reaction with a second molecule of isocyanide prior to reaction with TMS azide to form a tetrazole product. The partition between (1) and (3) is controlled by temperature with transformation (3) occurring at -40 to -20 degrees over long reaction times, and (1) occurs in good yields and selectivities at 30 degrees. The processes that are involved are either 3- or 4-component reactions.

The feature of this manuscript that is most significant is the potential of the isocyanide-alkylidene adduct for transformations beyond those reported in this manuscript, and with the discovery that the magnesium(II)-N,N'-dioxide ligand is appropriate for high enantiocontrol this manuscript can be expected to stimulate further research in the area. I can envision successful outcomes from reactions with indole, with diazo compounds, including cycloaddition reactions, and, perhaps, even with vinyl ether. This manuscript has used the authors' well known catalysts, the increasingly common concept of multicomponent reactions, and mildly reactive and easily handled materials to formulate complex chemical processes that are intrinsically dependent on the alkylidene-malonate, isocyanide, and catalyst as the foundation for new chemical syntheses.

There are a few grammatical errors that the authors will want to correct: (1) change "catalytical" to "catalytic"; (2) pg 1, right column, l 10: I do not understand the meaning of "potentially anticipates in" - change "in" to "the"?; (3) 3rd paragraph before Conclusion, 10 lines from bottom: "is intrigued by" is not a suitable selection of words; (4) 3rd paragraph before Conclusion, 9 lines from bottom: "equal" instead of "equival".

Michael P. Doyle

Reviewer #1 (Remarks to the Author):

In this manuscript, Liu, Feng and coworkers report an interesting isocyanide-based asymmetric multicomponent reaction that allows for the synthesis of nitrogen-embedded heterocycles such as tetrazoles and 1,2-dihydroisoquinolines with high enantiomeric purity. Critical to the success is the unprecedented initiation involving chiral MgII-N,N'-dioxide catalyst-enabled enantioselective addition of simple isocyanide to alkenes. By judicious adjustment of the reaction parameters, multicomponent reactions of isocyanides, TMSN₃ with alkylidene could be tuned in either three-component or four-component fashion. Moreover, the dearomative [3+2] cycloaddition of nonactivated isoquinolines could be achieved based on the same Zwitterionic intermediate for the first time. All these reactions constitute the extremely rare successful examples in controlling the enantioselectivity for multicomponent reactions. Mechanistic studies also well supported the proposed catalytic cycle. In addition, the SI is prepared in high quality.

Therefore, this reviewer suggests the publication of these highly interesting results in Nat. Commun. after the following minor points being addressed.

- 1) Abstract, line 3: "relying on..." should be changed to "relying on..."
- 2) Highlight of Scheme 1: "High efficient with..." should be changed to "Highly efficient with..."
- 3) In the footnote of Table 1, 1 and 3 should be changed to 1a, 3a in ref g.
- 4) Page 5, the last sentence: "claissic Passerini and Ugi..." should be changed to "classic Passerini and Ugi..."
- 5) Page 6, footnote of Table 4: "All reaction were carried out 7..." should be changed to "All reactions were carried out with 7..."
- 6) Page 7, line 8, left column: "23% yield" should be changed to "21% yield" (according to Table 4)?
- 7) Page 8, line 23, left column: "equivale" should be changed to "equivalent".

Reply to the comments by reviewer 1

1. **Question:** Abstract, line 3: "relying on..." should be changed to "relying on...".
Response: We have changed it.
2. **Question:** Highlight of Scheme 1: "High efficient with..." should be changed to "Highly efficient with...".
Response: We have corrected the sentence.
3. **Question:** In the footnote of Table 1, **1** and **3** should be changed to **1a**, **3a** in ref g.
Response: We have corrected it.
4. **Question:** Page 5, the last sentence: "claissic Passerini and Ugi..." should be changed to "classic Passerini and Ugi...".
Response: We have corrected the sentence.
5. **Question:** Page 6, footnote of Table 4: "All reaction were carried out 7..." should be changed to "All reactions were carried out with 7...".
Response: We have corrected the sentence.
6. **Question:** Page 7, line 8, left column: "23% yield" should be changed to "21% yield" (according to Table 4)?
Response: We have corrected it. We reconfirm the yield is 21% according our original

experimental record book.

7. **Question:** Page 8, line 23, left column: "equivale" should be changed to "equivalent".

Response: We have changed to "equal" according another reviewer's suggestions, and we think the two words have the same meanings.

Reviewer #2 (Remarks to the Author):

Through the work described in this manuscript, Liu, Feng and co-workers have solved a very important and challenging problem in asymmetric catalysis. As the authors rightly discussed in the introduction, development of catalytic enantioselective variant of isocyanide-based multicomponent reactions has remained a prominent challenge and only a handful of reports exist till date.

Here, the authors have devised a very smart strategy to generate a common intermediate through enantioselective conjugate addition of isocyanides to alkylidene malonates using magnesium-complex of the versatile N-oxide ligand developed in their own lab. The common intermediate is then utilized for three different reactions. In the first two of these reactions, azide addition followed by annulation take place to form a tetrazole. An impressive aspect is that, simply by modifying the reaction conditions, the involvement of either one or two molecules of isocyanide in this reaction can be controlled. In the last reaction, the common intermediate acts as a 1,3-dipole and participates in a formal [3+2]-cycloaddition reaction with isoquinoline.

While the substrate scope is quite wide, the enantioselectivities are not excellent in all the cases. However, this can be ignored considering the challenge at hand and the novelty of the strategy.

With the help of a number of control experiments, a reasonable catalytic cycle is proposed along with a stereochemical model to explain the observed stereoselectivity.

Overall, the work reported in this manuscript makes a very important contribution to the field of organic synthesis. Therefore, I would like to recommend the publication of this manuscript in Nature Communications.

Following minor issues should be sorted out before the manuscript is accepted for publication:

- (1) It would be nice to have a couple of applications and synthetic elaborations of the products.
- (2) In a few places in the first page, 'participate' is mistakenly written as 'anticipate'. These and similar mistakes should be corrected.

Reply to the comments by reviewer 2

1. **Suggestion:** It would be nice to have a couple of applications and synthetic elaborations of the products.

Response: We have tried some transformations. For 1,2-dihydroisoquinoline derivatives, we cannot reduce the C=C bond and C=N bond by Pd-C and H₂. However, we can get the

5-NH₂ substituted product. It indicated the 1,2-dihydroisoquinoline skeleton is more stable. We also failed to remove the ester and imide groups. For tetrazoles, we also examined the imide and tetrazole skeletons. We find the imide is stable and difficult to be reduced and oxidized. When we were successful to remove the ester group in high temperature, the imide group still exists (h). Finally, we added the transformations (a) to the supporting information.

2. **Question:** In a few places in the first page, ‘participate’ is mistakenly written as ‘anticipate’. These and similar mistakes should be corrected.

Response: We have corrected it.

Reviewer #3 (Remarks to the Author):

The significance of this manuscript is captured in Scheme 2. Coordination of the alkylidene-malonate by an appropriate chiral-ligated Lewis acid activates the substrate for highly enantioselective Michael addition by isocyanides to produce an intermediate that is now capable

of multiple transformations. The authors have identified three transformations that incorporate this intermediate - (1) reaction with TMS azide to form chiral tetrazoles, (2) reaction with isoquinolines to produce chiral heterocycles by dearomatization, and (3) reaction with a second molecule of isocyanide prior to reaction with TMS azide to form a tetrazole product. The partition between (1) and (3) is controlled by temperature with transformation (3) occurring at -40 to -20 degrees over long reaction times, and (1) occurs in good yields and selectivities at 30 degrees. The processes that are involved are either 3- or 4-component reactions.

The feature of this manuscript that is most significant is the potential of the isocyanide-alkylidene adduct for transformations beyond those reported in this manuscript, and with the discovery that the magnesium(II)-N,N'-dioxide ligand is appropriate for high enantiocontrol this manuscript can be expected to stimulate further research in the area. I can envision successful outcomes from reactions with indole, with diazo compounds, including cycloaddition reactions, and, perhaps, even with vinyl ether. This manuscript has used the authors' well known catalysts, the increasingly common concept of multicomponent reactions, and mildly reactive and easily handled materials to formulate complex chemical processes that are intrinsically dependent on the alkylidene-malonate, isocyanide, and catalyst as the foundation for new chemical syntheses.

There are a few grammatical errors that the authors will want to correct: (1) change "catalytical" to "catalytic"; (2) pg 1, right column, l 10: I do not understand the meaning of "potentially anticipates in" - change "in" to "the"; (3) 3rd paragraph before Conclusion, 10 lines from bottom: "is intrigued by" is not a suitable selection of words; (4) 3rd paragraph before Conclusion, 9 lines from bottom: "equal" instead of "equivale".

Michael P. Doyle

Reply to the comments by reviewer 3

- 1. Question:** Change "catalytical" to "catalytic"
Response: We have changed it.
- 2. Question:** pg 1, right column, l 10: I do not understand the meaning of "potentially anticipates in" - change "in" to "the"?
Response: We have changed "anticipates" to "participates".
- 3. Question:** 3rd paragraph before Conclusion, 10 lines from bottom: "is intrigued by" is not a suitable selection of words.
Response: We have changed "intrigued" to "triggered".
- 4. Question:** 3rd paragraph before Conclusion, 9 lines from bottom: "equal" instead of "equivale".
Response: We have corrected it.